# Bioabsorbable Polymeric Stent for the Treatment of Coarctation of the Aorta (CoA) in Children: A Methodology to Evaluate the Design and Mechanical Properties of PLA Polymer

**DOI:** 10.3390/ma16124403

**Published:** 2023-06-15

**Authors:** Flávio José dos Santos, Bruno Agostinho Hernandez, Rosana Santos, Marcel Machado, Mateus Souza, Edson A. Capello Sousa, Aron Andrade

**Affiliations:** 1Department of Mechanical Engineering, Centre for Simulation in Bioengineering, Biomechanics and Biomaterials, School of Engineering (CS3B), Campus of Bauru, UNESP—São Paulo State University, São Paulo 17033-360, Brazil; bruno.agostinho@unesp.br (B.A.H.); marcel.bergamo@unesp.br (M.M.); mateus.p.souza@unesp.br (M.S.); edson.capello@unesp.br (E.A.C.S.); 2Department of Engineering, PUC—Pontifical Catholic University of São Paulo, São Paulo 05014-901, Brazil; rosana@pucsp.br; 3CEAC—Centre for Engineering in Circulatory Assistance, Dante Pazzanese Institute of Cardiology, São Paulo 04012-909, Brazil; aandrade@fajbio.com.br

**Keywords:** bioabsorbable stent, coarctation of the aorta, finite element analysis

## Abstract

This study presents a methodology that combines experimental tests and the finite element method, which is able to analyse the influence of the geometry on the mechanical behaviour of stents made of bioabsorbable polymer PLA (PolyLactic Acid) during their expansion in the treatment of coarctation of the aorta (CoA). Tensile tests with standardized specimen samples were conducted to determine the properties of a 3D-printed PLA. A finite element model of a new stent prototype was generated from CAD files. A rigid cylinder simulating the expansion balloon was also created to simulate the stent opening performance. A tensile test with 3D-printed customized stent specimens was performed to validate the FE stent model. Stent performance was evaluated in terms of elastic return, recoil, and stress levels. The 3D-printed PLA presented an elastic modulus of 1.5 GPa and a yield strength of 30.6 MPa, lower than non-3D-printed PLA. It can also be inferred that crimping had little effect on stent circular recoil performance, as the difference between the two scenarios was on average 1.81%. For an expansion of diameters ranging from 12 mm to 15 mm, as the maximum opening diameter increases, the recoil levels decrease, ranging from 10 to 16.75% within the reported range. These results point out the importance of testing the 3D-printed PLA under the conditions of using it to access its material properties; the results also indicate that the crimping process could be disregarded in simulations to obtain fast results with lower computational cost and that new proposed stent geometry made of PLA might be suitable for use in CoA treatments—the approach that has not been applied before. The next steps will be to simulate the opening of an aorta vessel using this geometry.

## 1. Introduction

Coarctation of the aorta (CoA) is a congenital heart disease (CHD) in which a stenosis on the aorta artery occurs. In other words, there is a narrowing in the aorta obstructing the flow of blood. It accounts for 6–8% of all congenital heart diseases [1,2,3,4].

Among the consequences of CoA, the following can be listed: high blood pressure, breath shortness, tiredness when performing physical exercises, an increase in the pressure difference between the extremities (greater than 20 mmHg), tingling of the extremities, poor limb development, and, in extreme occasions, brain aneurysms [3,5,6]. Early identification of CoA is therefore essential for patient survival. According to Brown et al. [4], the lack of adequate treatment for CoA in the early years can increase the mortality rate by up to 80% in patients up to 50 years of age.

Studies have presented an incidence of CoA ranging from 1 in 2500 [4] to 1 in 2900 live births [3] in the mid-2010s. With an estimated worldwide annual birth rate of around 150 million births [7], approximately 51,000 to 60,0000 newborns per year worldwide would have some degree of coarctation in the aorta. The early discovery and treatment of CoA are therefore essential for the child’s complete motor-physiological development. The treatment of patients with CoA consists of restoring the diameter of the aorta, i.e., widening the section with coarctation. One of the most common treatments for CoA is the use of metallic stents [1]. Studies have shown that the use of stents in CoA has the highest success rate among all available treatment techniques: approximately 98% of cases did not present any serious complications [6], and injuries to the aortic wall were not reported (compared with the balloon angioplasty technique—10% of the cases), although stent fractures were observed in 22% of the cases, without consequences for the patients [3]. However, the simulations carried out by Pathirana et al. [8] suggest that the resection and end-to-end anastomosis treatment may result in fewer long-term complications than metallic stent treatments of CoA.

The use of metallic stent in newborns or small children, however, is not recommended: as the child grows, the diameter of the aorta would also increase, loosening the stent and allowing its movement inside the artery, leading to thrombosis. Moreover, the stent would have to be re-expanded, increasing the plastic deformation of the stent structure and increasing the likelihood of fracture [9,10]. According to Kasar et al. [11], surgical repair is the gold standard treatment for CoA in infants and young children. Nevertheless, stent implantation remains a controversial strategy in infants below 15 kg due to relatively large sheath dimensions as well as the inability to accommodate adult vessel sizes after growth [12].

An alternative to metallic stents, and a viable option for use in children and newborns, is the use of bioabsorbable stents, usually made of polymers, specifically PLLA (Poly-L-Lactide Acid) and PLA (PolyLactic Acid) [13]. As these polymers have smaller mechanical strength and rigidity than metals, the main task of this type of stent is to provide sufficient structural strength to support the forces and pressures from the vessels and arteries using only the stent’s geometry architecture [14,15]. This is particularly challenging in the aorta vessel as this structure is much larger and stiffer than the coronary vessel, which would require additional strength from the bioabsorbable stent to expand and hold the contraction forces. Consequently, this is the main reason for the non-existence, to the authors’ knowledge, of bioabsorbable stents made of polymeric material for the treatment of CoA in children up to this moment. This was also pointed out by Veeram et al. [16], who stated that at present there are no biodegradable stents available for use in paediatric patients with congenital heart disease despite the large occurrence of CoA in children. The use of bioresorbable alternatives to metal stents would eliminate the extra cost and risk associated with follow-up revascularization procedures needed to force permanent implants to match patient growth [17].

Numerical methods, especially the finite element method (FEM), have been widely used in the analysis of new cardiovascular equipment and the development and optimization of new stents [18]. This method allows the analysis of structures with nonlinear material properties, with complex loading and boundary conditions, to access a full field of stress and strain levels, and it has a relatively low cost when compared with traditional experimental procedures [19]. However, despite its popularity, the majority of the models available in the literature are either designed to analyse stents for coronary arteries or are not focused on the structural analysis of the stent for the treatment of coarctation of the aorta [18]. More specifically, structural and design analysis of bioabsorbable stents is rarely found [13].

One of the steps during surgery for stent insertion is crimping. In this procedure, the stent is compressed into a smaller diameter to be inserted into the catheter: this generates initial plastic deformation and introduces residual stresses. Schiavone et al. [20] compared the mechanical performance of a metallic and a polymeric stent for application in coronary arteries using finite element models. Comparisons were conducted in terms of radial expansion and stress levels arising from stent/artery contact. The finite element simulations also considered the crimping process and the residual stresses arising from this process. The authors concluded that crimping did not alter the stress distribution during the implantation process and only imposed small changes in the stress magnitudes. To the authors’ knowledge, there are no published studies evaluating the effect of crimping on the mechanical performance of polymeric stents for the treatment of aortic coarctation.

Another important factor in analysing the performance of a new stent is the ability to expand without recoil. Qiu et al. [13] studied the biomechanical behaviour during the crimping and expansion of four types of commercial polymeric bioabsorbable stents used in coronary artery repair. In this study, three-dimensional finite element models of the four types of stents were created and computationally subjected to the conditions of crimping and expansion. All stents were evaluated in terms of von Mises stress and recoil (elastic recoil). It was observed that in all cases the recoil was higher when the residual stresses from the crimping were included in the model. To the authors’ knowledge, there are no published studies assessing von Mises stress and recoil levels in polymeric stents for the treatment of aortic coarctation.

Due to the problems related to metallic stents and their limitations in paediatric use for the treatment of CoA, the relative occurrence of coarctation in the aorta in young children, and the lack of studies on the structural performance of bioabsorbable stents for the treatment of CoA, this study aimed to develop a methodology that combined experimental testing and the finite element method to analyse the influence of geometry on the mechanical behaviour of a bioabsorbable 3D-printed stent developed by the Centre for Engineering in Circulatory Assistance (CEAC), the Dante Pazzanese Institute of Cardiology (IDPC), during its expansion for the treatment of CoA. This model was developed based on the original prototype and its performance was evaluated in terms of linear-recoil, recoil (elastic recoil), and stress levels.

## 2. Materials and Methods

Figure 1 presents this study’s workflow.

### 2.1. Stent Geometry

The stent geometry was developed by the Centre for Engineering in Circulatory Assistance (CEAC), the Dante Pazzanese Institute of Cardiology (IDPC). The geometry was made available in CAD format and then prepared in the SpaceClaim software (v2022R2, Ansys Inc., Canonsburg, PA, USA). The stent had a total length of 25 mm, 11 rings in the axial direction, bar-shaped connecting elements, an external diameter of 6.75 mm, and a thickness of 0.25 mm, as shown in Figure 2a. The stent’s circular perimeter is illustrated in Figure 2b.

### 2.2. Determination of the Mechanical Properties of PLA

Fifteen (n = 15) Type II standard specimens, according to ASTM D638-14, Standard Test Method for Tensile Properties of Plastics [21], made of PLA material (3D Fila Indústria e Comércio Limitada, Belo Horizonte, Brazil) were 3D-printed on a Hadron Max 3D printer (Version V1, Wietech Industria, Osasco, Brazil), with a resolution of 50 microns on X and Y axis and 25 microns Z axis; mechanical precision of 50 microns on X and Y axis and 10 microns on the Z axis; a filament diameter of 1.75 mm; layer height of 0.2 mm; the printing speed of 33 mm/s; perimeter speed outer surface of 31 mm/s and filling speed of 31 mm/s. The extruder temperature was 190 °C and the table temperature was 60 °C. The printer nozzle diameter was 0.4 mm. Figure 3 shows a Type II standard specimen.

A tensile test was conducted using a Kratos testing machine (resolution of 0.01 mm, model K2000MP, Kratos Equipamentos Industriais Ltda., Cotia, Brazil), with a maximum loading capacity of 2000 kg, a load cell of 2000 kgf, and a loading speed of 5 mm/min.

### 2.3. Model’s Validation Experiment

A tensile test to later validate the stent model was performed using five (n = 5) customized 3D-printed PLA stent specimens in a planar condition at a scale of 3.5:1, as shown in Figure 4. The PLA material, printing, and testing parameters were the same as described in Section 2.2. At the beginning of the experiment, the initial stent circumferential length $\left(L_{0}\right)$, i.e., the stent’s circular perimeter as shown in Figure 2b, was measured by a digital calliper (precision of 0.01 mm, model 798 A-6/160, Starret Indústria e Comércio Ltda., Itu, Brazil). A displacement loading was then applied elongating the stent up to 78% of $L_{0}$, reaching the maximum value of $L_{max}$. This elongation percentage is equivalent to the expansion of the stent to a diameter of 12 mm.

In order to measure the linear-recoil effect of the stent, the maximum stent length (*L_max_*) at maximum displacement (78% of $L_{0}$) was measured for each sample using a digital calliper (precision of 0.01 mm, model 798 A-6/160, Starret Indústria e Comércio Ltda., Itu, Brazil). Sequentially, after loading removal, the final length of each sample ($L_{f}$) was also acquired using the same digital calliper. The linear-recoil effect of the stent was calculated using the equation:(1)$$linear-recoil=\frac{L_{max}-L_{f}}{L_{max}}$$
where $L_{max}$ is the maximum length and $\;L_{f}$ is the final length of the geometry.

### 2.4. Finite Element Modelling

In this study, two finite element models were developed using Ansys software (v2022R2, Ansys Inc., Canonsburg, PA, USA). The first one aimed to simulate the validation experiment described in Section 2.3, Figure 4, while the second one simulated the effect of crimping on the mechanical performance of the stent as well as the expansion performance of the stent.

A mesh convergence study for the stent was performed using the model without crimping, varying the sizes of the stent elements from 0.16 mm to 0.1 mm, as shown in Table 1. All models were expanded to 12 mm. All other boundary conditions remained the same as in the main analysis. With an error of 1.19% in the maximum von Mises stress, a stent element size of 0.1 mm was assumed. As the crimping and balloon cylinders were defined as rigid bodies, linear elements were used. For the stent geometry, quadratic elements were used to improve accuracy. The use of at least two elements in the stent cross section was assured. More details are presented in the next section.

#### 2.4.1. Finite Element Modelling of Experimental Tests

A first model was developed to simulate the expansion of the stent in the planar condition, during the customized tensile test to validate the stent model, see Figure 5. One side of the stent geometry was fixed, and a displacement of 78% of its initial length $\left(L_{0}\right)$ was applied on the other side, in the same way as described in Section 2.3. The mechanical properties of the PLA used in the simulation were those experimentally found in Section 2.2. Boundary conditions are shown in Figure 5a and the type of element used is shown in Table 2.

#### 2.4.2. Finite Element Modelling of Crimping

A second model was developed to simulate the crimping of the stent from its initial diameter of 6.75 mm to 5 mm and its subsequent expansion to a maximum diameter that ranged from 12 mm to 15 mm. These simulations were performed according to the simplifying hypotheses presented by Donik et al. [22], which considered the application of a displacement on a rigid surface, the crimper, to simulate the crimping process and later the application of a displacement by another rigid surface, the balloon, to expand the stent diameter to the maximum value. For the crimper, the steel material available in the Ansys software library was used, with a modulus of elasticity of 200 GPa, a Poisson’s ratio of 0.3, and a yield point of 250 MPa. For the balloon, a hyperelastic material was used according to the second-order Mooney Rivlin equation [23], with constants $C_{10}=1.03$, $C_{01}=3.69$ and $D_{1}=0$. Details of the boundary conditions are shown in Figure 5b. All materials were considered homogeneous and isotropic [24].

A 3D solid element with a quadratic interpolation function, ten nodes, and three degrees of freedom per node was used for the stent. For the crimper and the balloon, shell elements were used, with four nodes and six degrees of freedom at each node. Table 2 shows the final mesh details and element types. Contact elements between the stent and balloon and the crimper and the stent were used with a friction coefficient of 0.05 [25,26] and 0.8 [20], respectively.

In order to simulate the surgical procedures of inserting a stent into the aorta artery, the following steps were performed:(1)A radial displacement by the rigid crimper was applied to reduce the stent’s initial diameter from 6.75 mm to 5 mm.(2)The crimper was removed so that the stent could go through the linear-recoil effect.(3)The stent was expanded from 5 mm to a maximum value that ranged from 12 to 15 mm by applying a radial displacement by the balloon.(4)The balloon was removed to measure the elastic recoil.

## 3. Results

### 3.1. Mechanical Properties of PLA

The results obtained by the tensile test with standardized specimens according to Section 2.2 were treated using Excel software (Microsoft Office 365, Microsoft Inc., Redmond, WA, USA), and analysed using Minitab software (v21.3.1, Minitab LLC, Philadelphia, PA, USA). The obtained curve was divided into two parts: the first part was the elastic region of deformation while the second one was the plastic region.

For the construction of the elastic region, the initial nonlinear part of the experiment was not considered as it was related to the initial accommodation of the testing machine. Therefore, only the straight segment between the displacement of 1 mm to 3.7 mm was considered, Figure 6a. As can be seen in Figure 6a, the correlation between the applied load (N) and displacement (mm) is strong with an R^2^ of 94.2%.

To find the elastic modulus of the PLA, an average curve from all force vs displacement curves of Figure 6a was generated and transformed into a stress vs strain curve using specimens’ geometrical data, as shown in Figure 6b, obtaining a modulus of elasticity of 1.5 GPa. The yield point was 30.66 MPa. The same Poisson ratio as adopted by Qiu et al. [13] for PLA was chosen, approximately 0.3.

### 3.2. Finite Element Model Validation

The linear-recoil effect was calculated by Equation (1) and the experimental results are shown in Table 3. As the stent’s perimeter, in a 3.5:1 scale, was 73 mm, $\left(L_{0}\right)$, an average displacement of 56.8 mm was applied in both the experiment and the model, which was equivalent to a stent expansion to a diameter of 12 mm, representing about 78% of $L_{0}$. The average maximum experimental stent length, *L_max_*, was 130.02 mm. The finite element model presented a maximum displacement of 129.8 mm, a difference of 0.16%. The average experimental linear recoil was 36.36%, with an average final stent length, *L_f_*, of 82.74 mm. The model presented a linear recoil of 33.76%, with a final displacement, *L_f_*, of 12.98 mm (Figure 7), reaching a final length of 85.98 mm—a difference of 3.48% compared with the experimental results.

### 3.3. Assessment of Crimping Effect on Stent’s Mechanical Performance

According to Section 2.4.2, the stent was initially crimped from its initial diameter of 6 mm to a smaller diameter of 5 mm and then expanded to a maximum diameter that ranged from 12 mm to 15 mm. To compare the effect of crimping in stent mechanical performance during expansion, the recoil effect was once again calculated, with and without the crimping process, and Equation (1) was modified as follows:(2)$$recoil\left(\%\right)=\frac{d_{max}-d_{f}}{d_{max}}\times 100$$
where $d_{max}$ is the maximum diameter and $\;d_{f}$ is the final stent diameter measured in the centre of the stent.

The von Mises equivalent stress was evaluated to analyse the effect of crimping on the mechanical performance of the stent. Four scenarios were assessed as presented in Figure 8: (1) after the crimping process, when the stent reached its smallest diameter; (2) after the removal of the crimper, during the pre-recoil; (3) during stent maximum expansion to a diameter of 12 mm; and (4) after the balloon withdrawal.

As can be seen in Figure 9a, the maximum stress levels during the crimping process were around 34.84 MPa, which decreased to around 16.50 MPa after crimping, see Figure 9b. The stress levels were increased again during the stent expansion to a diameter of 12 mm (Figure 9c), reaching a maximum of 35.37 MPa. After removing the balloon (Figure 9d), the maximum stress levels were reduced and stabilised to around 30.66 MPa.

Figure 10 pictures the stent expansion without crimping. The von Mises stress levels when the stent was expanded to a diameter of 12 mm, as shown in Figure 10a, reached a maximum value of 37.43 MPa, decreasing to 30.57 MPa after the balloon was removed.

## 4. Discussion

Coarctation of the aorta (CoA) is a heart disease in which the aorta artery diameter is abruptly reduced in the aorta arc section, decreasing the blood flow and causing motor-physiological underdevelopment of the child [1,2,3,4]. The identification of CoA at early ages is thus essential for patient survival [4].

One of the main treatment options for CoA is the use of metallic stents [1,6]. However, the use of traditional metallic stents in newborns or small children is not recommended due to the children’s constant growth [9,10]. An alternative to metallic stents is the use of bioabsorbable stents, usually made of polymers, specifically PLA (PolyLactic Acid) [13]. Nonetheless, different from the ones made for the coronary vessels, the material strength of such polymers is not enough to support the aorta artery contraction forces. As a result, the main challenge in this type of stent is to provide enough structural strength to withstand such forces using the stent’s geometry architecture [14,15].

One of the crucial steps of the stent insertion surgery is the crimping procedure. In order to insert the stent in the catheter, the stent undergoes extensive compressive loading and deformations. Therefore, any stent project must analyse the stent performance during this procedure. There are several studies in the literature evaluating the crimping behaviour of stents for coronary vessels [13,20]. However, to the authors’ knowledge, this analysis was not yet conducted for polymeric stents for the treatment of aortic coarctation. Moreover, to assess the new stent’s mechanical performance, it is necessary to analyse von Mises stresses and recoil levels, which, to the authors’ knowledge, was also never conducted for this kind of device and treatment.

Consequently, this study has aimed to develop a methodology that combines experimental testing and the finite element method to analyse the influence of geometry and the crimping procedure on the mechanical behaviour of a new bioabsorbable 3D-printed stent developed by the Dante Pazzanese Institute of Cardiology (IDPC) during the stent’s expansion for the treatment of CoA.

Initially, the 3D-printed PLA material was mechanically characterised by standardised tensile tests. The 3D-printed PLA presented an elastic modulus of 1.5 GPa and a yield strength of 30.6 MPa. In a study conducted by Donik et al. [22], a finite element model of a bioabsorbable PLA stent was developed, and 3.0 GPa and 65.0 MPa were assigned for the elastic modulus and yield strength, respectively, for the PLA material. These values are much higher than those obtained in the current study. However, Donik et al. [22] referenced their properties to an extruded PLA, not 3D-printed as in this study. Several factors might have influenced the mechanical properties of the 3D-printed PLA polymer, such as printing speed and direction. In addition, the loading rate in the tensile test can also influence these properties. This study used ASTM D638-14, the Standard Test Method for Tensile Properties of Plastics, to correctly set the loading speed for such polymers. Nonetheless, further studies should be conducted to better assess the influence of these parameters on the mechanical properties of 3D-printed PLA.

Sequentially, a validation experiment of the stent finite element model was performed via a tensile test using customized 3D-printed stent specimens. According to Forbes et al. [27], the ideal stent for treating CoA should be expandable from 12 to 22 mm in diameter, with small foreshortening, high fracture resistance, and adequate radial resistance at maximum diameter. The experimental linear recoil in this study was calculated as 36.36%, while the numerical model presented the 33.76% linear recoil, thus indicating a good correlation between the experimental data and the numerical model.

In terms of the influence of crimping on the mechanical performance of the stent, Figure 8 shows that crimping has a small influence on stent circular recoil, with an average difference of 1.81% for expansion of diameters ranging from 12 mm to 15 mm. The stent expansion to maximum diameters ranging from 12 mm and 14 mm was to simulate the opening of an aorta vessel in children aged 85 to 180 months old [28]. The expansion to 15 mm had the purpose of analysing the stent deformation capacity for larger vessels. Similar results were found by Qiu et al. [13], who studied the biomechanical behaviour of four types of bioabsorbable polymeric stents for coronary arteries during crimping and expansion. It was observed by the authors that in all cases the recoil was marginally higher when the residual stresses from the crimping were included in the model and the average recoil was around 17%, similar to the current study. In a study carried out by Donik et al. [22], the authors analysed the recoil performance of stents made of two bioabsorbable materials, PLA and PCL (polycaprolactone). They found that recoil levels ranged from 8% to 26%, similar to this study.

Regarding the equivalent von Mises stress levels for the maximum expansion with crimping (Figure 9c), and without crimping (Figure 10a), a difference of 2.06 MPa was observed, indicating a greater plastic deformation compared with the non-crimped condition and consequently a lower recoil. In terms of the tensile stress after the balloon was removed, as shown in Figure 9d and Figure 10b, there was no change in stress levels.

Bioresorbable stents offer an attractive alternative to commercially available metal stents [17], but specific guidelines for paediatric coarctation treatments are lacking [2]. According to Veeram et al. [16], there is an urgent need for bioabsorbable stents for children with congenital heart disease (CHD), and hopefully, with advancements in technology, researchers and medical device companies will be able to manufacture such devices.

In terms of the study´s limitations, the coefficient of friction was taken from a study of a coronary artery and not from the aorta due to the lack of such data for the aorta vessel. The balloon was modelled as a rigid surface rather than a flexible balloon, but that approach is widely accepted in the field as it introduces minimal errors and saves computational resources. Another limitation was a comparison of the finite element model’s circular recoil with the experimental linear recoil. As the experimentation procedures for stents are quite complex and data acquisition is difficult, an experiment with customized stent specimens in planar condition rather than in the natural cylinder format was proposed. It was assumed that the circular contraction (recoil) was similar to the linear contraction of the stent. Finally, it was assumed that all materials were homogeneous and isotropic; for 3D-printed specimens, some variation in the mechanical properties was expected.

## 5. Conclusions

This study developed a methodology combining experimental testing and the finite element to analyse the influence of geometry on the mechanical behaviour of a bioabsorbable stent made of PLA during expansion for the treatment of CoA in children. The 3D-printed PLA was mechanically characterized using a tensile test, obtaining an elastic modulus of 1.5 GPa and a yield strength of 30.6 MPa, which is different from the raw material. With a validated model, it was possible to observe that the crimping process did not affect the stent´s mechanical performance during expansion as the recoil marginally increased; for expansion of diameters ranging from 12 mm to 15 mm, as the maximum opening diameter increased, the recoil levels decreased, ranging from 10 to 16.75%, which suggested that the new proposed geometry made in PLA might be suitable for use in CoA treatments since such studies were not conducted before. Future studies will aim to assess other mechanical properties of the stent, such as foreshortening, inserting an aorta geometry with coarctation in the study, and evaluate the stent’s performance in opening a coarcted aorta.

## Figures and Tables

**Figure 1 materials-16-04403-f001:**
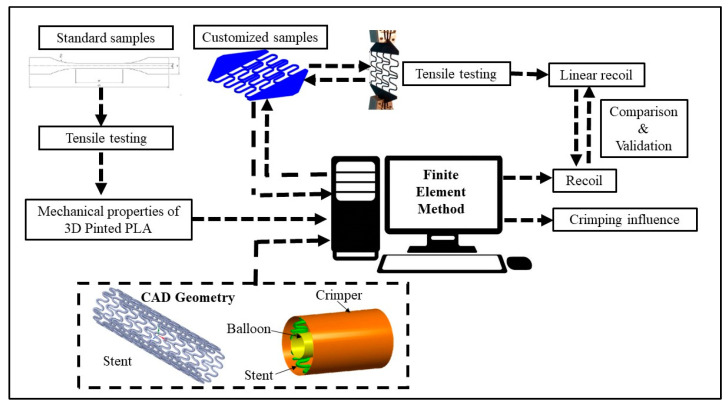
General workflow.

**Figure 2 materials-16-04403-f002:**
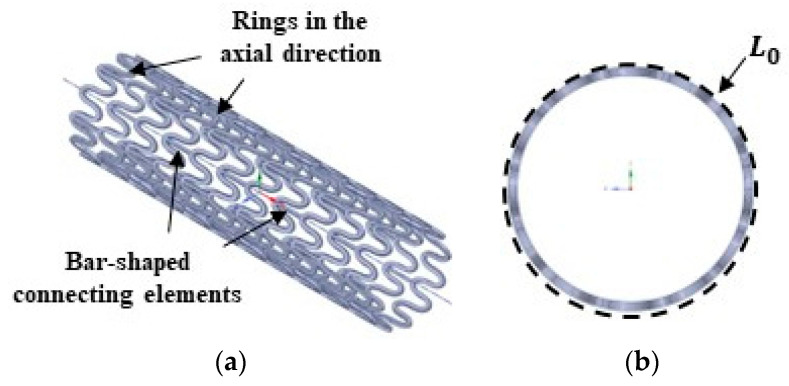
Stent geometrical features: (**a**) stent geometry and (**b**) $L_{0}$ circular perimeter.

**Figure 3 materials-16-04403-f003:**
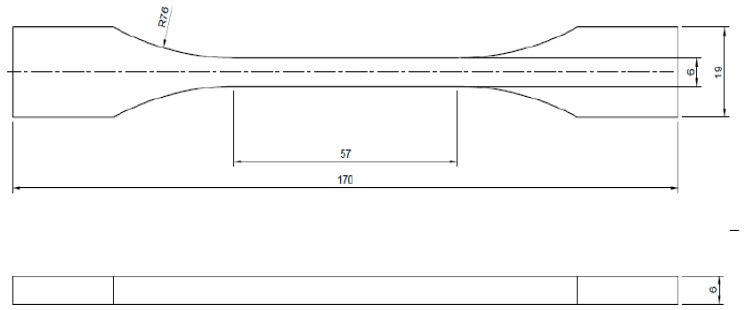
Type II standard specimen according to ASTM D638-14.

**Figure 4 materials-16-04403-f004:**
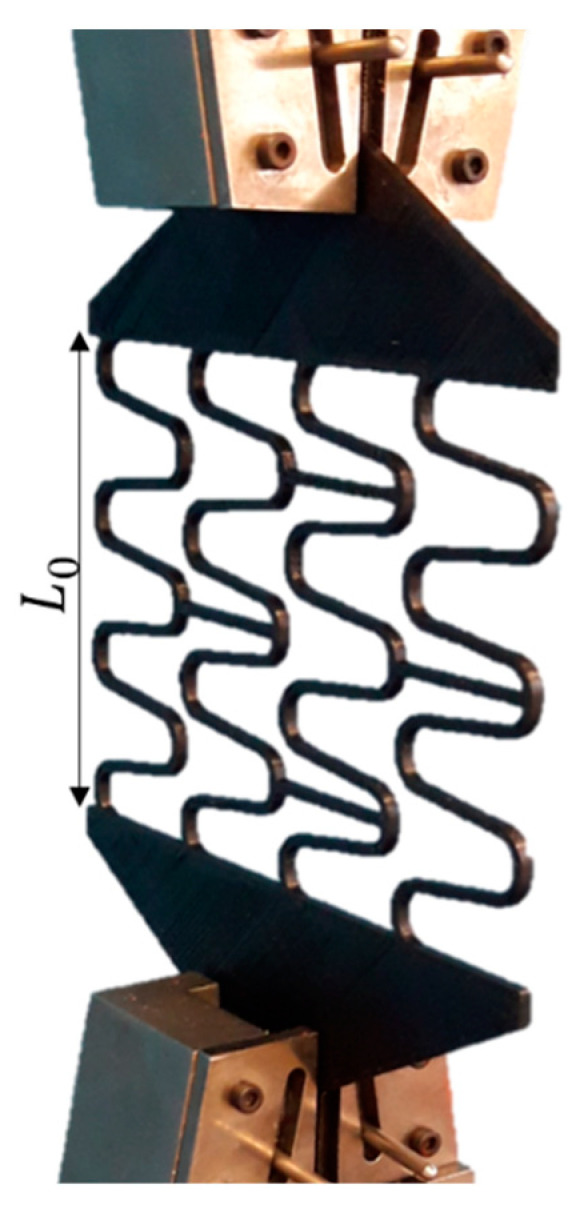
Customized 3D-printed PLA stent specimens in a planar condition at a scale of 3.5:1.

**Figure 5 materials-16-04403-f005:**
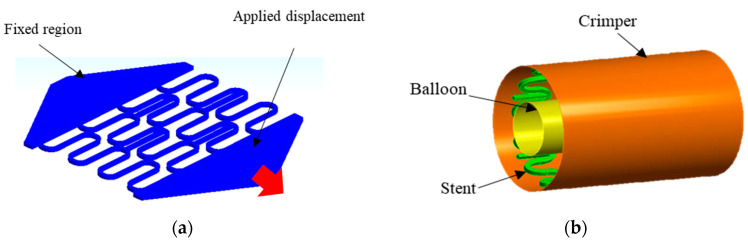
Finite element model boundary conditions: (**a**) boundary conditions for model 1 and (**b**) boundary conditions for model 2.

**Figure 6 materials-16-04403-f006:**
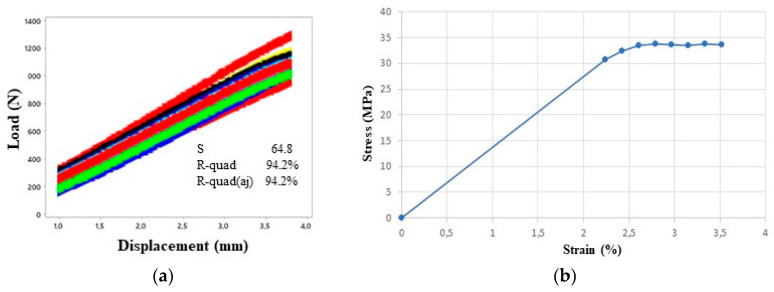
Tensile test. (**a**) Elastic region and (**b**) Stress vs strain curve.

**Figure 7 materials-16-04403-f007:**
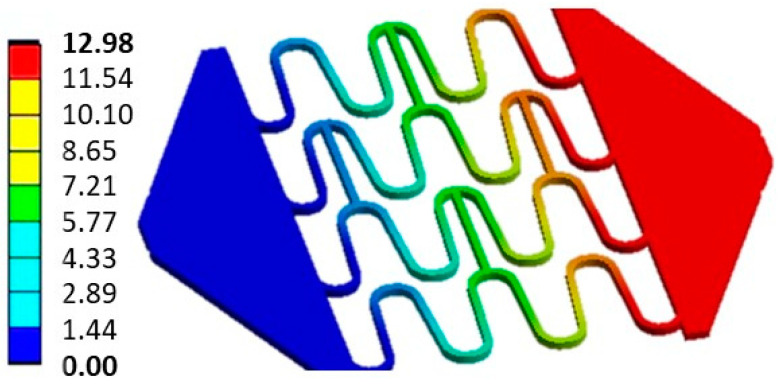
Finite element simulation of customized 3D-printed PLA stent specimens. Displacement in millimetres.

**Figure 8 materials-16-04403-f008:**
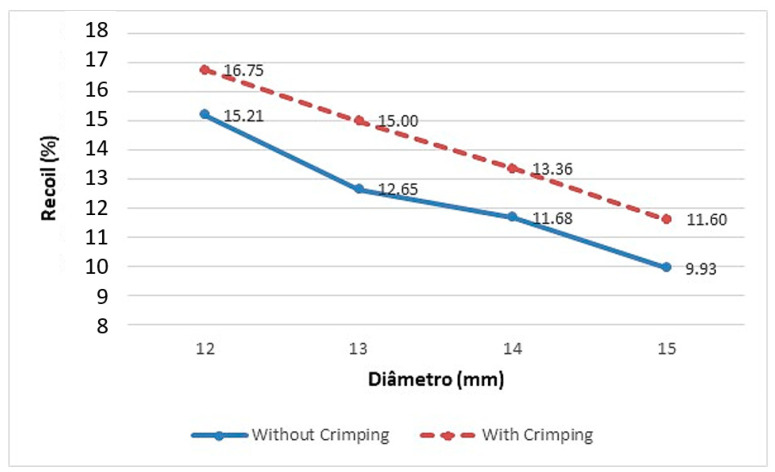
Assessment of crimping effect on the stent´s mechanical performance during expansion.

**Figure 9 materials-16-04403-f009:**
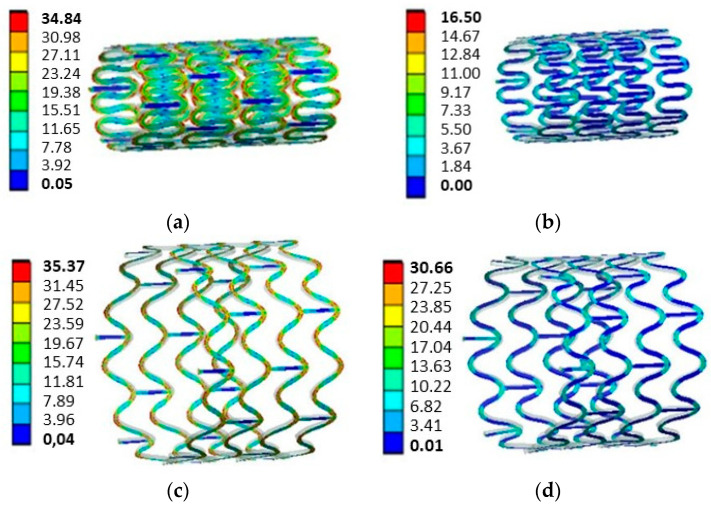
Stent von Mises stress levels in MPa: (**a**) after the crimping process, when the stent reaches its smallest diameter; (**b**) after the removal of the crimper, during the pre-recoil; (**c**) during stent maximum expansion to a diameter of 12 mm; and (**d**) after balloon withdrawal.

**Figure 10 materials-16-04403-f010:**
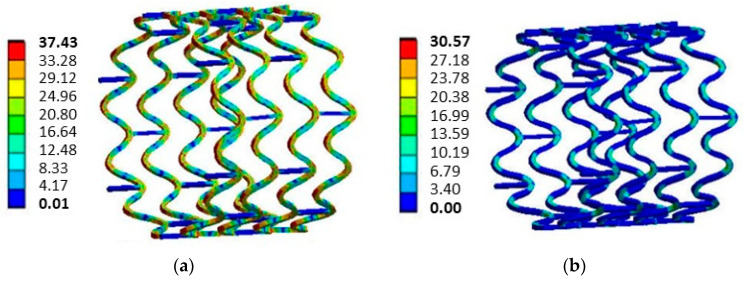
Stent von Mises stress levels without crimping in MPa: (**a**) the stent is expanded to a diameter of 12 mm and (**b**) after the balloon is removed.

**Table 1 materials-16-04403-t001:** Mesh convergence analysis.

Simulation	Element Size (mm)	Von Mises (MPa)	Error (%)
S1	0.16	40.34	-
S2	0.14	38.53	4.48
S3	0.12	37.88	1.70
S4	0.10	37.43	1.19

**Table 2 materials-16-04403-t002:** Mesh properties of the finite element model.

Part	Element Hype	Element Order	Number of Elements
Planned Stent	SOLID187	Quadratic	36,428
Stent	SOLID187	Quadratic	153,430
Crimper	SHELL181	Linear	1540
Balloon	SHELL181	Linear	672

**Table 3 materials-16-04403-t003:** Tensile test results—customized specimens.

Sample	*L*_0_ (mm)	*L_max_* (mm)	*L_f_* (mm)	*L_f_* − *L*_0_	Linear Recoil (%)	Load (N)
1	73.21	130.00	81.54	8.33	37.28	44.13
2	73.23	129.92	81.73	8.50	37.09	38.00
3	73.18	130.07	80.66	7.48	37.99	45.36
4	73.21	130.00	84.75	11.54	34.81	30.65
5	73.29	130.10	85.03	11.74	34.64	31.87
Average	73.22	130.02	82.74	9.52	36.36	38.00
Standard deviation	0.04	0.07	2.00	1.97	1.53	6.77

## Data Availability

Data will be made available as requested by anybody.

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
