# Peer review of "Bioabsorbable Polymeric Stent for the Treatment of Coarctation of the Aorta (CoA) in Children: A Methodology to Evaluate the Design and Mechanical Properties of PLA Polymer"

_materials, 2023, doi:10.3390/ma16124403_

Round 1

Reviewer 1 Report

This paper presents a well-structured study that aims to analyze the mechanical behavior of a bioabsorbable stent made of PLA during its expansion for the treatment of Coarctation of the Aorta (CoA) in children. The methodology presented in this study involves a combination of experimental tests and the finite element method to evaluate the influence of geometry on stent performance.

The authors began by characterizing the mechanical properties of 3D-printed PLA using tensile tests. This is a crucial step in understanding the material behavior and validating the finite element model. The finite element model was then generated from CAD files provided by CEAC - Dante Pazzanese Institute of Cardiology, and a rigid cylinder was created to simulate the stent opening performance. The authors also developed customized specimens to experimentally validate the finite element model.

The stent's performance was evaluated in terms of elastic return, recoil, and stress levels. The authors observed that 3D-printed PLA has a lower elastic modulus than pure PLA and metals. They also noted that crimping had little effect on the mechanical behavior of polymeric stents, and as the maximum opening diameter increases, the recoil levels decrease. This finding is important for the safety of the stent and suggests that the crimping process could be disregarded in simulations to obtain faster results and with lower computational costs in preliminary studies.

Overall, this study developed and validated a methodology combining numerical and experimental methods for the analysis of bioabsorbable stents for the treatment of CoA in children. The authors were able to demonstrate that their methodology is effective in analyzing the influence of geometry on the mechanical behavior of a bioabsorbable stent made of PLA during expansion. The study's findings contribute to the field of biomedical engineering by providing valuable insights into the design and performance of bioabsorbable stents for the treatment of CoA in children.

Author Response

Reviewer: 1

Comments to the Author:

Q: “English language and style are fine/minor spell check required.”

A: Thanks for the insightful comments. The paper was entirely revised, and grammar and typo mistakes were corrected. 

Reviewer 2 Report

In this article, a novel polymeric stent has been developed to treat the coarctation of the aorta in children. It is an innovative approach, as metallic stents should not be utilised in children and this pathology is relatively frequent in newborns. As much as I agree with the novelty of the device, and the fact that the authors have used biodegradable polymers, safety has not been tested. Do the authors have any data involving the safety of the device? This should be addressed before the article is accepted for publication in Materials.

There are also some minor comments:

·         Line 44: if the authors want to use CoA as an abbreviation of coarctation of the aorta, they should not us COA here. Please check

·         Figure 1: what does it mean ‘L0 perimeter’? It’s not clear what figure b means

·         Section 2.2: what does it mean ‘ASTM D638-14’?

Author Response

Reviewer: 2

Comments to the Author:

Q1: “In this article, a novel polymeric stent has been developed to treat the coarctation of the aorta in children. It is an innovative approach, as metallic stents should not be utilised in children and this pathology is relatively frequent in newborns. As much as I agree with the novelty of the device, and the fact that the authors have used biodegradable polymers, safety has not been tested. Do the authors have any data involving the safety of the device? This should be addressed before the article is accepted for publication in Materials.”

A: Thanks for the insightful comment. This paper represents an initial study to assess the material properties of 3D - printed PLA, strategies of modelling and to evaluate the design of a novel polymeric PLA stent to treat the coarctation of the aorta in children. The next step is to insert the geometry of an aorta with stenosis into the model and assess whether it supports the vessel.

Q2: “Line 44: if the authors want to use CoA as an abbreviation of coarctation of the aorta, they should not use COA here. Please check.”

A: Thanks for the insightful comment. The mistakes have been corrected.

Q3: “Figure 1: what does it mean ‘  perimeter’? It’s not clear what figure b means.”

A: Thanks for the insightful comment.  is a circular perimeter of the stent. The word “circular” has been added to the text.

Q4: “Section 2.2: what does it mean ‘ASTM D638-14’?”

A: Thanks for the insightful comment. ASTM D638-14 is a standardized test created by ASTM International which describes an experimental method test to access the tensile properties of plastics, published in 2014. This information has been added to the text.

Reviewer 3 Report

The present paper considers both physical and mathematical analysis of the behavior of stents manufactured by the additive method from PLA.

1. I recommend a better explanation of the bibliographic references in the introduction, see Lines 41, 50.

2. I recommend better contrast and increased text size in Figures where appropriate (see Figure 1, 5).

Author Response

Reviewer: 3

Comments to the Author

“The present paper considers both physical and mathematical analysis of the behavior of stents manufactured by the additive method from PLA.”

Q1: “I recommend a better explanation of the bibliographic references in the introduction, see Lines 41, 50.”

A: Thanks for the insightful comment. A better explanation of the text was added.

Q2: “I recommend better contrast and increased text size in Figures where appropriate (see Figures 1, 5).”

A: Thanks for the insightful comment. The figures have been improved.

Reviewer 4 Report

1.      Quantitative results need to be added in the abstract section.

2.      Given the "take-home" message at the end of the abstract, the present form was insufficient.

3.      Rearrange keywords alphabetically.

4.      Describe the novelty of the article made by the author? From the results of my evaluation, it seems that many similar published works adequately explain what you have raised in the current manuscript. If there is something others really new in this manuscript, please highlight it more clearly in the introduction section.

5.      The work, novelty, and constraints of relevant previous literature must be explained in the introduction section to highlight the article gaps that the present work aims to fill.

6.      It is suggested to the authors to make the objective o

7.      In line 64-70, the authors explain related to finite element method. In needs to discuss the advantages of computational simulation approach compared to experimental and clinical study such as lower cost and faster results. For this purpose, refer the relevant reference as follows: Polycrystalline Diamond as a Potential Material for the Hard-on-Hard Bearing of Total Hip Prosthesis: Von Mises Stress Analysis. Biomedicines 2023, 11, 951. https://doi.org/10.3390/biomedicines11030951

8.      Where is validation with proposer experimental results of the present study? It needs to shows.

9.      Meshing strategy needs to explained in the present computational simulation.

Author Response

Reviewer: 4

Comments to the Author

Q1: “Quantitative results need to be added in the abstract section.”

A: Thanks for the insightful comment. The main quantitative results have been added in the abstract section.

Q2: “Given the "take-home" message at the end of the abstract, the present form was insufficient.”

A: Thanks for the insightful comment. The “take-home” has been added in the abstract section. In summary, the results pointed out the importance of testing the 3D-printed PLA in the conditions of use to access its material properties, that the crimping process could be disregarded in simulations to obtain fast results with lower computational cost, and that the new proposed geometry made in PLA might be suitable to be used in CoA treatments, which approach was never applied.

Q3: “Rearrange keywords alphabetically.”

A: Thanks for the insightful comment. The keywords are already in alphabetic order in the paper.

Q4: “Describe the novelty of the article made by the author. From the results of my evaluation, it seems that many similar published works adequately explain what you have raised in the current manuscript. If there is something new in this manuscript, please highlight it more clearly in the introduction section.”

A: Thanks for the insightful comment. New pieces of information were added in the introduction and discussion sections to describe the novelty of the study.

Q5: “The work, novelty, and constraints of relevant previous literature must be explained in the introduction section to highlight the article gaps that the present work aims to fill.”

A: Thanks for the insightful comment. New pieces of information were added in the introduction and discussion sections to describe the novelty of the study.

Q6: “It is suggested to the authors to make the objective o.”

A: Thank you for the comment. Unfortunately, the comment was cut off and we couldn't fully understand it. Could you kindly rephrase it?

Q7: “lines 64-70, the authors explain related to the finite element method. It needs to discuss the advantages of the computational simulation approach compared to experimental and clinical studies such as lower cost and faster results. For this purpose, refer to the relevant reference as follows: Polycrystalline Diamond as a Potential Material for the Hard-on-Hard Bearing of Total Hip Prosthesis: Von Mises Stress Analysis. Biomedicines 2023, 11, 951. https://doi.org/10.3390/biomedicines11030951”

A: Thanks for the insightful comment. The suggested reference was analysed and added to the study´s scope. More information about the finite element method was also added.

Q8: “Where is validation with the proposer experimental results of the present study? It needs to show.”

A: Thanks for the insightful comment. This study described two experimental procedures. The first was about tensile testing of standard samples to determine the mechanical properties of the 3D – printed PLA (section 2.2). These results are presented in section 3.1. In the second experiment, we aimed to validate the stent finite element model through a tensile test using customized 3D-printed PLA stent specimens in a planar condition (section 2.3). These results are presented in section 3.2.

Q9: “The meshing strategy needs to explain in the present computational simulation.”

A: Thanks for the insightful comment. Mesh details have been added to section 2.4.

Reviewer 5 Report

The reviewer has few comments on this article:

Line 69-70-If this is research or the PLA, P-PLA material used for making newborn is rare, the authors should mention few researches on this, citing few names of researchers.

Line 106-208-Try to relate the sizes/dimensions stated in the lines to the paediatric size aorta. 

Line 262-297, the content is redundant with the introduction.

Line 299-302, the authors need to mention clearly whether this results come from this study or not.

For the conclusion-It is better to highlight the application of PA as newborn stents and also to include the best results in the conclusion.

Author Response

Reviewer: 5

Comments to the Author

Q1: “Line 69-70-If this is research or the PLA, P-PLA material used for making newborns is rare, the authors should mention a few pieces of research on this, citing a few names of researchers.”

A: Thanks for the insightful comment. To the authors knowledge there are a few studies regarding bioabsorbable stent for coronary disease, such as Qiu et al (14) and Schiavone et al (19). For bioabsorbable stents used for CoA in children, on the other hand, not a single study was found.

Qiu TY, Song M, Zhao LG. A computational study of crimping and expansion of bioresorbable polymeric stents. Mech Time-Dependent Mater. 2018; 22(2):273–90.

Schiavone A, Qiu TY, Zhao LG. Crimping and deployment of metallic and polymeric stents - finite element modelling. Vessel Plus. 2017; 1(1):12–21.

Q2: “Line 106-208-Try to relate the sizes/dimensions stated in the lines to the paediatric size aorta.”

A: Thanks for the insightful comment. This information was added in the discussion section.

Q3: “Line 262-297, the content is redundant with the introduction.”

A: Thanks for the insightful comment. The content was rewritten.

Q4: “Line 299-302, the authors need to mention clearly whether these results come from this study or not.”

A: Thanks for the insightful comment. The content was rewritten.

Q5: “For the conclusion-It is better to highlight the application of PLA as newborn stents and also to include the best results in the conclusion.”

A: Thanks for the insightful comment. The conclusion section was rewritten according to the comment.

Round 2

Reviewer 4 Report

Following comments are given as follows:

1.      The authors needs to explain the present work become more clear to understand.

2.      The reviewer encouraged the authors to provide an additional figure in the introduction section to improve the reader's understanding.

3.      To make the reader comprehend the workflow of the current study, the authors could include extra examples in the form of figures in the materials and methods rather than merely the dominating text as a present form.

4.      More information about tools, such as the producer, country, and specifications, should be included.

5.      Important information that must be included in the publication refers to the error and tolerance of the experimental equipment used in this inquiry.

6.      An evaluation of the findings with similar past investigations is required.

7.      Materials assumption is a crucial thing as input parameter consideration in computational simulation. It commonly used homogeneous, isotropic, and elastic as materials assumption. Please include this information along with the relevant reference as follows: Minimizing Risk of Failure from Ceramic-on-Ceramic Total Hip Prosthesis by Selecting Ceramic Materials Based on Tresca Stress. Sustain. 2022, 14, 1–12. https://doi.org/10.3390/su142013413

8.      Because the current quality is not appropriate, the authors must improve their discussion to add more depth.

9.      What is the current work's limitation? Please place it before entering the conclusion section.

10.   Write a paragraph-length conclusion rather than the existing form's point-by-point explanation.

11.   In the conclusion, please explain the further research.

12.   In the entire manuscript, the authors occasionally constructed paragraphs with just one or two phrases, which made the explanation difficult to understand. To make their explanation a full paragraph, the authors should expand it. It is advised to use at least three sentences in a paragraph, with the primary sentence coming first and the supporting sentences coming after.

13.   The reference needs to be enriched from the literature published five years back. MDPI reference is strongly recommended.

14.   The authors need to reduce their level of self-citation with using literature that not authored by the present authors in the current submission as a reference.

15.   Due to grammatical problems and linguistic style, the authors should proofread the work.

16.   Following the revision step, the authors must provide a graphical abstract.

Author Response

Response to the reviewers´ comments

The authors wish to thank the reviewers for their insightful comments. Our response to the reviewers is outlined below.

Reviewer: 4

Comments to the Author:

Q1: The authors need to explain the present work becomes more clear to understand.

A: Thanks for the comment. Unfortunately, we were not able to fully comprehend the question. Could you please further explain this to us? Which parts of the paper are not clear?

Q2. The reviewer encouraged the authors to provide an additional figure in the introduction section to improve the reader's understanding.

A: Thanks for the insightful comment. Which Figures would you suggest being added? A figure in the introduction is not a common practice in papers. The authors believe, as suggested by the other reviewers, that the figures in the materials and methods and results sections can sufficiently illustrate a coarcted aorta, the stent device, and the problem. 

Q3. To make the reader comprehend the workflow of the current study, the authors could include extra examples in the form of figures in the materials and methods rather than merely the dominating text as a present form.

A: Thanks for the insightful comment. A figure depicting the workflow was added to the text to improve readers' understanding of the study.

Q4. More information about tools, such as the producer, country, and specifications, should be included.

A: Thanks for the insightful comment. The authors have checked the whole paper and added missing information about the tools. The information is listed as follows:

- Hadron Max 3D printer (Version V1, Wietech Industria Ltda, Brazil), with a resolution of 50 Microns on the X and Y axis, and 25 Microns Z axis; Mechanical precision of 50 microns on the X and Y axis and 10 microns on the Z axis.

- Kratos Testing Machine (resolution of 0,01 mm, model K2000MP, Kratos Equipamentos Industriais Ltda, Brazil).

- Digital calliper (precision of 0.01 mm, model 798 A-6/160, Starret Indústria e Comércio Ltda, Brazil).

- Excel software (Microsoft Office 365, Microsoft Inc, USA).

- Minitab (v21.3.1, Minitab LLC, USA).

Q5. Important information that must be included in the publication refers to the error and tolerance of the experimental equipment used in this inquiry.

A: Thanks for the insightful comment. The information was added to the text and listed in the previous question.

Q6. An evaluation of the findings with similar past investigations is required.

A: Thanks for the insightful comment. As discussed in the paper, there is not a single study regarding bioabsorbable stents for the treatment of Aorta Coarctation in children. Consequently, we discussed and evaluated the results based on the past few studies that analysed stents for coronary diseases in adults and on some studies which evaluated the viability of stents for children, such as:

- Qiu TY, Song M, Zhao LG. A computational study of crimping and expansion of bioresorbable polymeric stents. Mech Time-Dependent Mater. 2018; 22(2):273–90.

- Schiavone A, Zhao LG. A study of balloon type, system constraint, and artery constitutive model used in finite element simulation of stent deployment. Mech Adv Mater Mod Process. 2015; 1(1):1.

- Schiavone A, Qiu TY, Zhao LG. Crimping and deployment of metallic and polymeric stents - finite element modelling. Vessel Plus. 2017; 1(1):12–21.

- Donik Ž, Nečemer B, Vesenjak M, Glodež S, Kramberger J. Computational Analysis of Mechanical Performance for Composite Polymer Biodegradable Stents. Mater (Basel, Switzerland). 2021; 14(20).

- Castaldi B, Ciarmoli E, Di Candia A, Sirico D, Tarantini G, Scattolin F, et al. Safety and efficacy of aortic coarctation stenting in children and adolescents. Int J Cardiol Congenit Hear Dis [Internet]. 2022;8(April):100389. Available from: https://doi.org/10.1016/j.ijcchd.2022.100389

- Veeram SR, Welch TR, Nugent AW. Biodegradable stent use for congenital heart disease. Prog Pediatr Cardiol [Internet]. 2021;61(2021):101349. Available from: https://doi.org/10.1016/j.ppedcard.2021.101349

- Wright J, Nguyen A, D’Souza N, Forbess JM, Nugent A, Reddy SRV, et al. Bioresorbable stent to manage congenital heart defects in children. Materialia. 2021 May 1;16:101078.

Q7. Materials assumption is a crucial thing as an input parameter consideration in computational simulation. It commonly used homogeneous, isotropic, and elastic as materials assumption. Please include this information along with the relevant reference as follows: Minimizing Risk of Failure from Ceramic-on-Ceramic Total Hip Prosthesis by Selecting Ceramic Materials Based on Tresca Stress. Sustain. 2022, 14, 1–12. https://doi.org/10.3390/su142013413

A: Thanks for the insightful comment. The material assumptions were included in the paper as well as the reference added.

Q8. Because the current quality is not appropriate, the authors must improve their discussion to add more depth.

A: Thanks for the insightful comment. As discussed in the paper, this paper aimed to introduce a preliminary study about a new stent device developed by CEAC Hospital for the treatment of Aorta Coarctation in children by analysing its initial mechanical performance. The lack of studies about the topic makes difficult a further and deeper analysis. Consequently, the authors think that the depth of the discussion, in agreement with the other four reviewers’ opinions, and with cardiac surgeons’ opinions from CEAC who read the paper, is adequate for the proposed initial analysis of this new device in this paper. Nonetheless, a couple of new thoughts were added to the discussion section.

Q9. What is the current work's limitation? Please place it before entering the conclusion section.

A: Thanks for the insightful comment. A paragraph describing work’s limitation was added to the Discussion section.

Q10. Write a paragraph-length conclusion rather than the existing form's point-by-point explanation.

A: Thanks for the insightful comment. The conclusion was rewritten.

Q11. In the conclusion, please explain further research.

A: Thanks for the insightful comment. The conclusion was rewritten.

Q12. In the entire manuscript, the authors occasionally constructed paragraphs with just one or two phrases, which made the explanation difficult to understand. To make their explanation a full paragraph, the authors should expand it. It is advised to use at least three sentences in a paragraph, with the primary sentence coming first and the supporting sentences coming after.

A: Thanks for the insightful comment. The paper was revised and modified as necessary.

  1. The reference needs to be enriched from the literature published five years back. MDPI reference is strongly recommended.

A: Thanks for the insightful comment. New references, from others and from MDPI, were added to the study as follows:

- Castaldi B, Ciarmoli E, Di Candia A, Sirico D, Tarantini G, Scattolin F, et al. Safety and efficacy of aortic coarctation stenting in children and adolescents. Int J Cardiol Congenit Hear Dis [Internet]. 2022;8(April):100389. Available from: https://doi.org/10.1016/j.ijcchd.2022.100389

- Homsi M, El Khoury M, Hmedeh C, Arabi M, El Rassi I, Bulbul Z, et al. Endovascular Stent Repair of Aortic Coarctation in a Developing Country: A Single-Center Experience. Cardiovasc Revasc Med. 2022 Jun;39:66–72.

- Kasar T, Erkut O, Tanidir IC, Şahin M, Topkarci MA, Guzeltas A. Balloon-expandable stents for native coarctation of the aorta in children and adolescents. Medicine (Baltimore). 2022 Dec;101(51):e32332.

- van Kalsbeek RJ, Krings GJ, Molenschot MMC, Breur JMPJ. Early and midterm outcomes of bare metal stenting in small children with recurrent aortic coarctation. EuroIntervention  J Eur Collab with Work Gr  Interv Cardiol Eur Soc Cardiol. 2021 Feb;16(15):e1281–7.

- Veeram SR, Welch TR, Nugent AW. Biodegradable stent use for congenital heart disease. Prog Pediatr Cardiol [Internet]. 2021;61(2021):101349. Available from: https://doi.org/10.1016/j.ppedcard.2021.101349

- Wright J, Nguyen A, D’Souza N, Forbess JM, Nugent A, Reddy SRV, et al. Bioresorbable stent to manage congenital heart defects in children. Materialia. 2021 May 1;16:101078.

- Ammarullah MI, Hartono R, Supriyono T, Santoso G, Sugiharto S, Permana MS. Polycrystalline Diamond as a Potential Material for the Hard-on-Hard Bearing of Total Hip Prosthesis: Von Mises Stress Analysis. Biomedicines [Internet]. 2023;11(3). Available from: https://www.mdpi.com/2227-9059/11/3/951

- Ammarullah MI, Santoso G, Sugiharto S, Supriyono T, Wibowo DB, Kurdi O, et al. Minimizing Risk of Failure from Ceramic-on-Ceramic Total Hip Prosthesis by Selecting Ceramic Materials Based on Tresca Stress. Sustainability [Internet]. 2022;14(20). Available from: https://www.mdpi.com/2071-1050/14/20/13413

  1. The authors need to reduce their level of self-citation by using literature that is not authored by the present authors in the current submission as a reference.

A: Thanks for the comment. Could you kindly point out where are the self-citations in the paper? We were not able to find them.

Q15. Due to grammatical problems and linguistic style, the authors should proofread the work.

A: Thanks for the insightful comment. Two colleagues, native English-British speakers from the University of Bath, UK, have read the paper and made a few adjustments to the text.

Q16. Following the revision step, the authors must provide a graphical abstract.

A: Thanks for the insightful comment. The graphical abstract was provided in the first submission. It will be uploaded again.
